# Atomic model for core modifying region of human fatty acid synthase in complex with Denifanstat

S. M. Naimul Hasan[1], Jennifer W. Lou [1], Alexander F. A. Keszei[2], David L. Dai[1] & Mohammad T. Mazhab-Jafari [1,2] ✉

Fatty acid synthase (FASN) catalyzes the de novo synthesis of palmitate, a 16-carbon chain fatty acid that is the primary precursor of lipid metabolism and an important intracellular signaling molecule. FASN is an attractive drug target in diabetes, cancer, fatty liver diseases, and viral infections. Here, we develop an engineered full-length human FASN (*h*FASN) that enables isolation of the condensing and modifying regions of the protein post-translation. The engineered protein enables electron cryo-microscopy (cryoEM) structure determination of the core modifying region of *h*FASN to 2.7 Å resolution. Examination of the dehydratase dimer within this region reveals that unlike its close homolog, porcine FASN, the catalytic cavity is close-ended and is accessible only through one opening in the vicinity of the active site. The core modifying region exhibits two major global conformational variabilities that describe long-range bending and twisting motions of the complex in solution. Finally, we solved the structure of this region bound to an anti-cancer drug, Denifanstat (*i.e.*, TVB-2640), demonstrating the utility of our approach as a platform for structure guided design of future *h*FASN small molecule inhibitors.

De novo fatty acid synthesis is a conserved multi-step reaction (Fig. 1A) found in bacteria and eukaryotes. Enzymes involved in the synthesis of fatty acids can be divided into two classes based on organization of the catalytic assembly. Type II fatty acid synthases (FAS) found in most bacteria are composed of individual functional proteins, each originating from distinct genes that collectively carry out the condensation and modification cycles of fatty acid synthesis[1]. Substrate shuttling and protein-protein interactions are governed by random diffusion. In contrast, Type I FAS found in eukaryotes and some actinobacteria are large complexes of multi-domain protomers that are composed of one or two repeating polypeptides that are each encoded by a single gene and organized in a symmetric or pseudo-symmetric assembly[2]. In humans, the *FASN* gene encodes a 273 kDa polypeptide that comprises the catalytic domains necessary for palmitate synthesis[3], which further homodimerizes into a pseudo-symmetric complex (Fig. 1B)[4].

*h*FASN is composed of two distinct structural regions each containing two copies of their constituent catalytic domains: 1) the condensing region−ketoacyl synthase (KS), linker domain (LD), and malonyl/acetyl transferase (MAT) domains; and 2) the modifying region−dehydratase (DH), pseudo-methyltransferase (ΨME), pseudo-ketoacyl reductase (ΨKR), enoyl reductase (ER), ketoacyl reductase (KR), acyl carrier protein (ACP), and thioesterase (TE) domains. ACP and TE domains are connected by a long flexible linker allowing for substrate shuttling across catalytic centers in the complex and final cleavage/release of palmitate, respectively. The contact point of the condensing and modifying regions is the neck region, a flexible linker composed of two short unstructured loops which allow for continuous swinging and swiveling motions of the two regions relative to each other, maximizing ACP accessibility to all catalytic centers[5].

*h*FASN is a validated anti-cancer target and is over-expressed in solid tumors including those from breast, prostate, liver, and lung

[1]Department of Medical Biophysics, University of Toronto, Toronto, Ontario, Canada. [2]Princess Margaret Cancer Center, University Health Network, Toronto, Ontario, Canada. ✉e-mail: mohammad.mazhabjafari@utoronto.ca

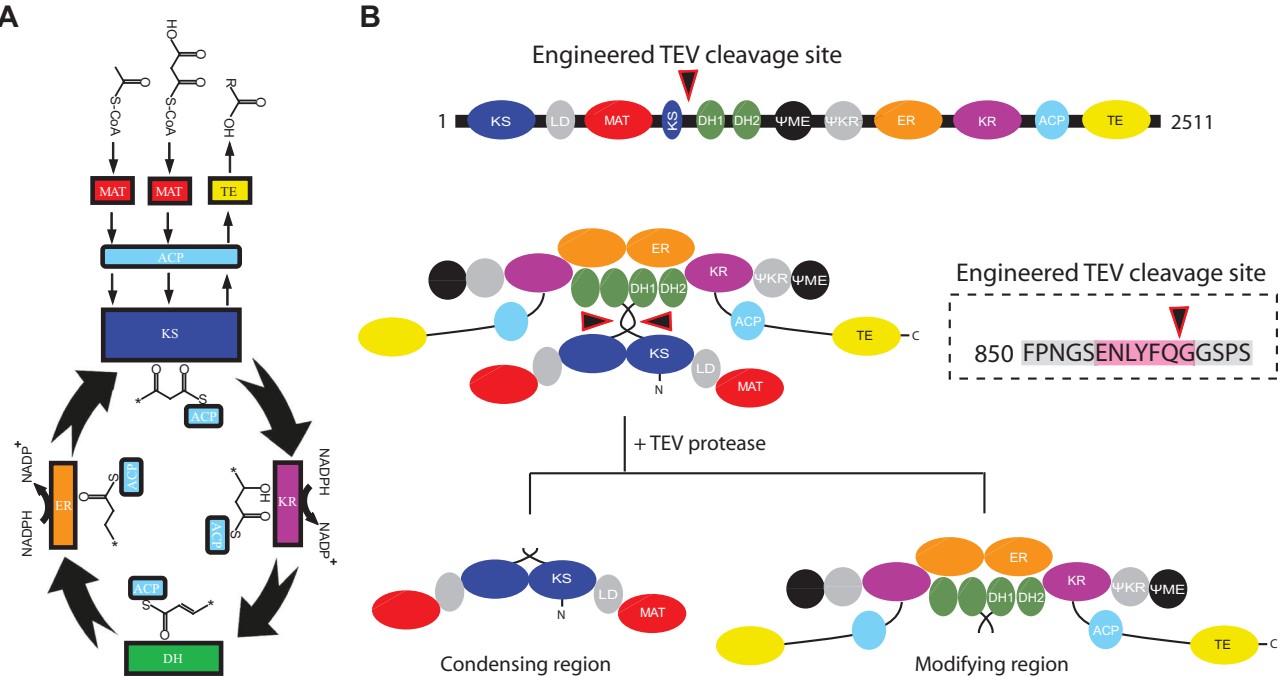

**Fig. 1 | Strategy for separation of the condensing and modifying regions of *h*FASN. A** Schematic representation of the catalytic cycle for de novo fatty acid synthesis in metazoans. **B** Domain organization and schematic of full length *h*FASN.

Location of the engineered TEV cleavage site is shown by the black-red arrowhead. The expected cleavage fragments are shown at the bottom of the panel.

tissues[6,7]. In breast cancer cell lines, such as HER2-positive SkBr3 and luminal A ZR75-1, 10–30% of cytosolic protein mass is composed of *h*FASN[8]. Despite remarkable efforts in structural and medicinal chemistry that have generated a repertoire of small molecule inhibitors against every catalytic domain of *h*FASN[9], only one—Denifanstat (TVB-2640), an imidazopyridine-based reversible KR inhibitor—has gone on to human clinical trials[10]. There is thus a need for rapid and efficient high-resolution structure determination of *h*FASN in complex with small molecule inhibitors to bolster drug screening and development efforts.

Mammalian FASN is a highly dynamic protein as judged via negative stain electron microscopy[5], cryoEM[11], atomic force microscopy[12], and molecular dynamic (MD) simulation[13] studies. The first cryoEM reconstruction of *h*FASN purified from breast cancer cell lines was reported in 2002[14]. It demonstrated significant heterogeneity in the quaternary organization of the full-length protein. This conformational heterogeneity was later corroborated for FASN from other mammalian species[5]. The crystal structure of porcine FASN revealed the underlying mechanism of conformational plasticity:[4] flexible linkers between well-structured regions of the complex enable independent movement of different parts of the complex, thereby maximizing the reach of the ACP domain to the catalytic centers. Specifically, three linkers in each protomer confer flexibility. The first is a 22 amino acid linker between the MAT and DH domains that define the boundary between the condensing and modifying regions. Based on the crystal structure of porcine FASN, the four C-terminal residues of the MAT-DH linker are predicted to be highly flexible while the remaining N-terminal residues are tightly associated with the KS homodimer and a linker domain sandwiched between the KS and MAT domains[4]. Indeed, swinging and swiveling motions pivoting at the MAT-DH linker[5] have been observed between the condensing and modifying regions of the complex. The other two flexible linkers are found at the KR-ACP and ACP-TE boundaries encompassing 12 and 26 amino acids, respectively. Both linkers are predicted to be unstructured and highly flexible as they have not been observed in

experimental electron density maps of porcine FASN and have large displacements in MD simulations of *h*FASN[13].

Currently, there are no atomic resolution experimental structures of full-length homodimeric *h*FASN. A key challenge in the structural determination of *h*FASN is the independent motions of its condensing and the modifying regions relative to each other. Previous studies have attempted to overcome this challenge by expressing truncations or individual domains of the protein. For example, expression and purification of an N-terminal fragment of *h*FASN spanning KS:LD:MAT domains, enabled structural studies of the condensing region bound to small molecule inhibitors and acyl-groups mimicking reaction intermediates[15]. The flexibly tethered TE domain has also been structurally characterized in isolation[16]. Furthermore, the ER domain[17], the ACP domain in complex with the human phosphopantetheinyl transferase (PPT transferase)[18], and an engineered ΨME:ΨKR:KR tridomain in complex with small molecule inhibitors[19] have been crystalized and provide the first snapshots of fragments of the *h*FASN modifying region. The domains of the modifying region are, in fact, targets of the majority of *h*FASN inhibitors to date[9] including the only clinically available anti-cancer drug, Denifanstat[15]. Nevertheless, a structure of the "core modifying region" of *h*FASN, hereafter defined as DH:ΨME:ΨKR:ER:KR domains, has not yet been solved to near atomic-resolution.

Here we report two protein preparation methods for *h*FASN modifying region that enable high resolution cryoEM studies. First, we demonstrate that insertion of a tobacco etch virus (TEV) protease cut site in the neck region of the *h*FASN complex enables cleavage of recombinantly purified proteins into isolated condensing and modifying regions. Excess molar ratio of TEV relative to FASN is required to achieve ~90% cleavage. Using this technique, we solved the structure of the core modifying region of *h*FASN and characterized its domain dynamics and interaction with the small molecule inhibitor Denifanstat. Second, we demonstrate that the modifying region of *h*FASN can be expressed and purified independently as a homodimer with superior yield and purity compared to the TEV cleavage method.

We discuss the potential advantage of each method in study of FASN structure.

## Results and discussion

### Isolation of condensing and modifying regions

Independent and continuous motions of the condensing and modifying regions generate multiple flexible FASN conformations that hamper structural determination of the entire complex to sub-nanometer resolutions by cryo-EM. Therefore, we rationalized that isolating the modifying region could result in a more conformationally stable protein amenable to cryo-EM structure determination. Truncation experiments on rat FASN demonstrated that the condensing region may be essential for dimerization of mammalian FASN as deletion of this region results in a monomeric protein[20]. This notion was supported by the crystal structure of porcine FASN as the KS homodimer forms the largest interface between the two protomers of the complex, (2580 Å²)[21]. However, a more recent truncation study on mouse FASN demonstrated successful expression and purification of a proteolytically stable fragment, corresponding to the modifying region of the mouse protein[22]. We therefore tested if optimal dimerization and assembly of the modifying region of *h*FASN may require formation of the N-terminal KS homodimer during mRNA translation. Implicit in our hypothesis is that once the modifying region is formed it no longer requires the condensing region for structural stability as evidenced by the free motions of the two regions independent of each other in the context of the full complex. Therefore, we first engineered a TEV cleavage site into the flexible and solvent accessible MAT-DH linker to separate the condensing and modifying regions post-translation. The protease cleavage site was inserted between a natural GSGS sequence found in the flexible region of the MAT-DH linker (Fig. 1B). Compared to full-length, the modified protein was purified to similar yield and displayed a similar elution profile by size-exclusion chromatography, suggesting insertion of the cleavage site does not impact folding & stability (Fig S1). We next attempted to cleave the protein and surprisingly found that the protease was needed in 200-fold excess molar ratio to achieve ~90% cleavage (Fig. 2A), perhaps due to steric hinderance of the cut site in certain conformations. The cleaved condensing and modifying regions were separated using size exclusion chromatography (Fig. 2B and C) as the hydrodynamic radius of the latter is approximately twice as large due to the flexibly attached ACP and TE domains.

CryoEM analysis of the modifying region revealed an arch shaped homodimer composed of two of each of the DH, ΨME, ΨKR, ER, and KR domains while lacking density for the C-terminal ACP & TE domains (Fig. 2C). We also observed trace amounts (~10%) of uncleaved *h*FASN in cryo screenings (Fig. 2B), demonstrating that insertion of the TEV cut site preserves the native fold of the full complex. Overall, these data demonstrated that it is possible to isolate and purify *h*FASN modifying region from the full complex post-translationally.

We next tested whether the modifying region of *h*FASN could be expressed and purified independently. We optimized the truncation boundary based on the crystal structure of porcine FASN[4] and placed it in the middle of the GSGS linker in the human FASN sequence. Prior truncation studies on rat FASN included a small portion of the condensing region[20] due to lack of high-resolution structural information at the time. To our surprise, *h*FASN modifying region was expressed and purified with a yield approximately ten times higher (Fig. 2D) than the TEV cleavage method (see method section for details). The preparation was at higher purity as judged via SDS-PAGE analysis (Fig. 2C, D). Increased protein yield was in part due to higher cell viability at the time of harvest indicating over-expressing full length *h*FASN was metabolically more costly for the HEK293F cells.

The experiments described above indicate that expression and purification of truncated *h*FASN modifying region results in superior yield and purity compared to proteolytic cleavage of the full-length protein. Higher yield and purity will be beneficial for downstream structural biology and drug development application of this isolated fragment. In contrast, the TEV cleavage method enables expression of full-length protein in cells, which is an important factor when studying *h*FASN structure from natural sources such as those purified from cancer cell lines[8,14]. Expression of truncated form of the multienzyme complex may be highly disruptive to the host cell metabolism and cellular signaling that regulate FASN protein function[23]. Furthermore, *h*FASN function in human cells is regulated via post translational modification (PTM) such as acetylation[24], phosphorylation[25], lactylation[26], O-GlcNAcylation[27], S-nitrosylation[28], and ubiquitination[29]. These PTMs depend on the cells metabolic status and cellular signaling networks that may be disrupted if truncated forms of *h*FASN are expressed. Proteolytic separation of *h*FASN condensing and modifying regions may then enable high-resolution investigation of naturally occurring PTMs of this protein when expressed under near physiological conditions. Using CRISPR gene editing technology a TEV cleavage site can be inserted into the endogenous *FASN* gene in cancer cell lines such as a luminal-A or a HER2 positive breast cancer cell line from which engineered *h*FASN can be purified without affinity tags[8,14]. Isolation of the functional modules of *h*FASN from these natural sources will then open a new avenue to study the conformations of the endogenous multienzyme complex at near atomic resolution. Additionally, the TEV cleavage method will enable structural studies of transient interactions within the modifying region of *h*FASN. Specifically, the substrate promiscuity of the MAT domain in the condensing region can be used to acylate the holo ACP domain in the modifying region using unusual acyl-CoAs mimicking FASN reaction intermediates[30]. Holo ACP domain can be generated enzymatically by PPT transferase enzymes such as Sfp and AcpS[31]. TEV mediated separation of the FASN condensing and modifying regions post-acylation will then enable cryoEM based studies of transient ACP interactions with inactivated catalytic domains in the core modifying region of the multienzyme complex (i.e., KR, DH, and ER domains). Catalytic inactivation is crucial to ensure that acylated ACP domains are stalled at the inactivated catalytic site. We have used a similar concept to construct the cryoEM structure of stalled ACP domain of *S. cerevisiae* FAS at the DH domain of the fungal multienzyme complex[32]. For these reasons we proceeded with structural studies of the core modifying region prepared from the TEV cleavage method.

### Structure and conformational variability

The isolated modifying region of *h*FASN produced via the TEV cleavage method demonstrated adequate orientation distribution to enable 3-dimensional reconstruction of a cryoEM map of the complex (Figs. S2 and S3) into which an atomic model for the core modifying region was built (Fig. 3). Many residues within the ΨME and ΨKR domains lacked defined side chain densities, although backbone density was clearly defined. This observation suggested the presence of significant conformational dynamics within the two pseudo domains at the periphery of the *h*FASN core modifying region.

No cryoEM densities were observed in the 3D reconstruction that could be assigned to the ACP and TE domains, consistent with the inherent flexibility of these domains relative to the core. Our structure of the human core modifying region (Supplementary Table 1) aligns well with that of porcine FASN (78% sequence identity) with a backbone RMSD of 1 Å (Supplementary Table 2), indicating its structural integrity in isolation. Surprisingly, close examination of the human DH domain revealed a close-ended catalytic cavity as opposed to the porcine homolog where DH domain has an open-ended cavity (Fig. 4A). This is due to the presence of a leucine residue (i.e., L1097) instead of an alanine at the back end of the catalytic cavity in the human protein (Fig. 4A).

As opposed to porcine, most metazoans have a bulky hydrophobic residue lining the back end of their DH catalytic cavities

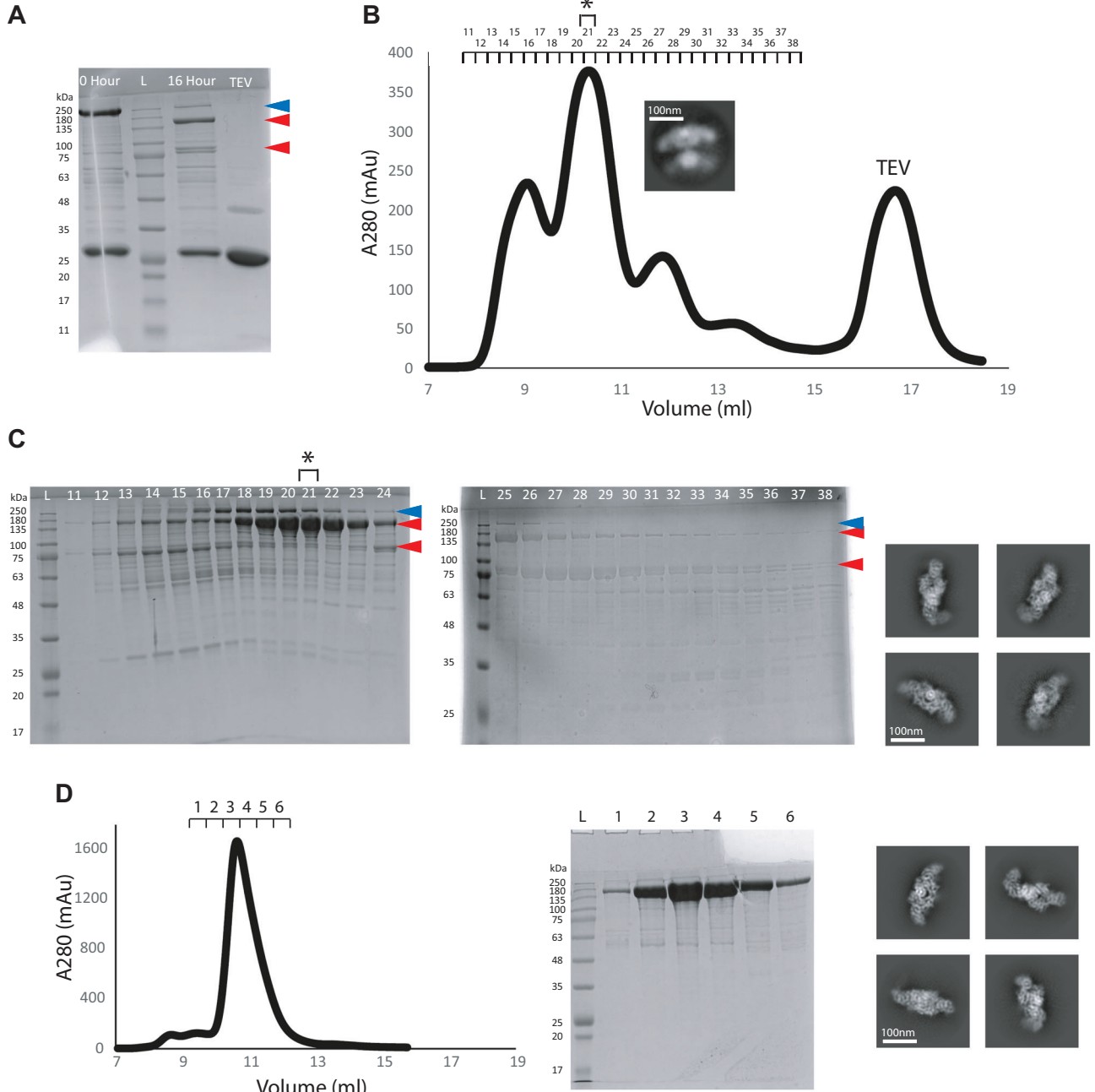

**Fig. 2 | Isolation of the modifying region of *h*FASN. A** SDS-PAGE analysis using Coomassie staining of the TEV cleavage reaction is shown. Blue and red arrows are full length and cleaved *h*FASN, respectively. **B** Separation of cleavage product using size exclusion chromatography. A representative 2D class of traces of uncleaved hFASN in fraction 21 is shown. **C** SDS-PAGE analysis of the elution fractions (same staining as in (**A**)). Representative 2D classes from particle images frozen in vitreous ice are shown from fraction 21 highlighted by *. **D** Elution profile of *h*FASN modifying region expressed and purified as an independent construct. SDS-PAGE analysis of elution fractions and 2D classes of fraction 3 are shown. Experiments from (**A**) and (**C**) were repeated once and **D** repeated two independent times.

(Fig. 4B), suggesting a close-ended cavity is a general feature for type I FAS in multicellular eukaryotes. We tested the impact of this natural structural variation on the catalytic function of the DH domain via a functional assay using 3-hydroxybutyryl-CoA and NADPH (Fig. S4). In this assay, MAT domain acylates holo ACP domain by transfer of the hydroxybutyryl group from coenzyme A to phosphopantetheine group of the shuttling domain[30,33]. The hydroxybutyryl-ACP is then a substrate for the DH domain that converts it to crotonyl-ACP in a reversible reaction[33]. Crotonyl-ACP is a substrate for the ER domain that utilizes the NADPH cofactor in a reduction reaction to produce the final product butyryl-ACP. Therefore, by monitoring NADPH consumption in this coupled assay, the function of the DH domain can be

deduced. It is important to note that not all the ACP domains may be phosphopantothenated since FASN is overexpressed in HEK293F cells. We observed no significant change in NADPH consumption by FASN upon L1097A mutation (Fig. S4) indicating that metazoan FASN proteins can well tolerate close- and open-ended DH catalytic cavities for hydroxybutyryl substrate. However, it is possible that longer acyl chains may impact the DH catalyzed reaction differently when the back end of the reaction cavity is closed vs open.

Next, we probed the dynamics of the core modifying region using a 3D variability analysis (3DVA) algorithm[34]. We probed for three principal components that described the variability within the data. This analysis identified two major continuous motions of the protein

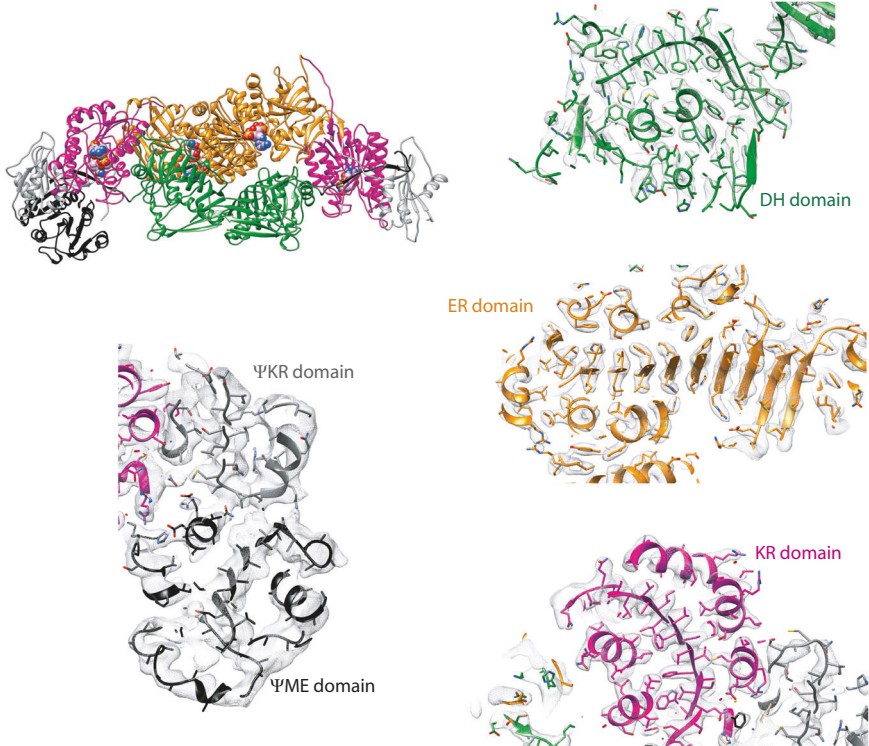

**Fig. 3 | Model to map fit of individual *h*FASN domains.** Slices of atomic model of *h*FASN fitted into the cryoEM density is shown for each domain. Sharpened map is used. Amino acid side chains within the pseudo domains were modeled as alanine when no corresponding densities were observed.

from its 2D particle projections. The first principal component, which represents the largest variability within the particle images, showed a bending motion of the core modifying region (supplementary Movie 1 and supplementary Movie 2) pivoted at the ER dimer interface. The second greatest source of variability in the dataset was explained by a twisting motion mediated via sliding of the DH domain against the ER homodimer (supplementary Movie 3 and supplementary Movie 4) accompanied by a displacement of the ΨME:ΨKR:KR trimer. The third most significant variability within the particle images was the appearance and disappearance of density corresponding to the ΨME:ΨKR heterodimer in our reconstructions (Supplementary Movie 5). This along with the previous observation indicate that there is significant conformational heterogeneity present in the pseudo domains of *h*FASN. We also found the catalytic sites of the DH, ER, and KR domains are minimally perturbed by these observed global motions. Interestingly, we also observed cryoEM density corresponding to a loop at the entrance of the KR catalytic cavity which appears and disappears in all three PCA components (Supplementary Movies 2, 4, and 5), indicating that it is adopting multiple conformations.

The dynamics identified by this analysis is likely an underestimated representation of the full conformational landscape of the core modifying region as only 14% of all particle images of the complex contributed to the final high-resolution reconstruction (Supplementary Table 1).

*h*FASN is a critical drug target in many disorders. In addition to obesity and cancer, *h*FASN has also been suggested to be an effective target to combat viral infections such as SARS-CoV-2[35]. Therefore, we tested whether our TEV cleavage methodology enables near atomic-resolution structure determination of small molecule drugs complexed with the core modifying region, which is the target site for many *h*FASN inhibitors[6]. For this effort, we selected Denifanstat, an orally bioactive molecule with antineoplastic activity. The human KR domain is active from our preparations as assessed by monitoring oxidation of NADPH using trans-1-Decalone as a KR specific substrate.

The KR domain within full-length TEV engineered construct and the truncated modifying region expressed and purified independently, demonstrate specific activities of $236 \pm 84.8$ and $480 \pm 115$ mU/mg, respectively, where one mU is defined as consumption of one nmol of NADPH per minute (Fig. 5A). The slight difference in specific activity is likely a reflection of difference in the level of purity between the two preparations (Fig. 2). Denifanstat (TVB-2640) blocks consumption of NADPH by the KR domain in both protein preparations (Fig. 5A). This drug is currently in use in multiple clinical trial against solid tumors[6] and non-alcoholic steatohepatitis[36]. A hydrophobic cavity in the ΨME:ΨKR:KR tridomain has been successfully targeted by small triazolone[19] and piperazine[37] compounds with masses of approximately 500 Daltons. This cavity extends from the catalytic core of the KR domain, which harbors the nicotinamide group of the NADPH cofactor, to the ΨME domain and is thought to be a binding pocket for ACP-linked acyl chains during fatty acid biosynthesis[19]. cryoEM map of the core modifying region of *h*FASN prepared from the TEV cleavage method in complex with Denifanstat was reconstructed to 2.6 Å resolution (Figs. S5 and S6 and Table S1). The global resolution of the drug bound complex was further improved by applying C2 symmetry to 2.4 Å resolution (Fig S7 and Table S1). We found that the drug binds the same pocket and interacts predominantly with catalytic residues from the KR domain and the nicotinamide of NADPH (Figs. 5B, C, S8). However, the binding conformation of Denifanstat is different from triazolone and piperazine compounds, with the Denifanstat being more solvent exposed and interacting with residues from the dynamic loop predicted to be involved in ACP binding (Fig. 5D). To the best of our knowledge, this is the first cryoEM model of *h*FASN complexed with the first-in-class anti-neoplastic inhibitor that is undergoing clinical trials and can be used as a platform for rational design of other novel *h*FASN inhibitors.

The approaches taken here, isolating *h*FASN by its functional regions, demonstrate a reliable path toward high-resolution structural studies of the core modifying region of the complex by cryo-EM. This

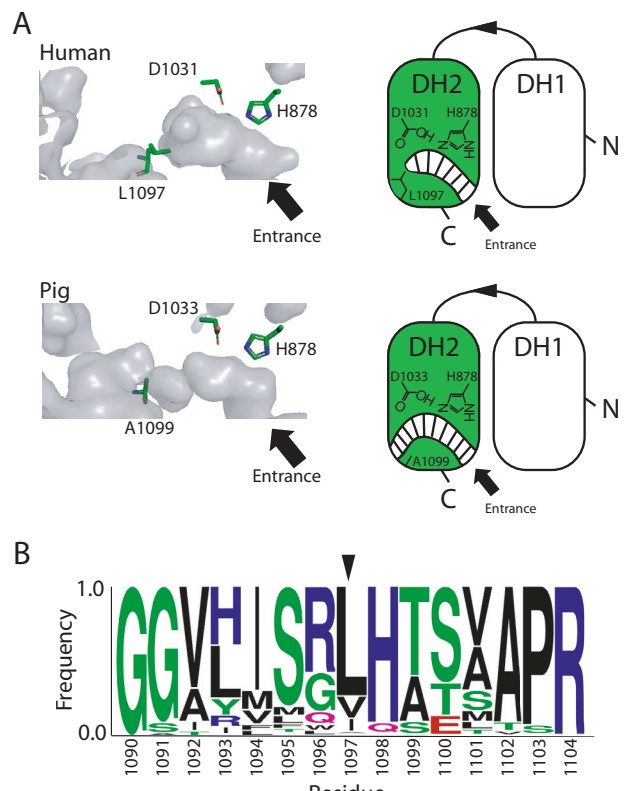

**Fig. 4 | Close-ended catalytic cavity of the human DH domain. A** Comparison of the cavity (shown as gray surface) harboring the catalytic dyad (shown as sticks) of the DH domain between human (top) and pig (*S. scrofa*, bottom) PDB 2VZ9. Schematic representations of the DH domains are shown to the right of each model. **B** Presence of bulky hydrophobic residue (indicated with arrow) at the back end of the DH catalytic cavity in metazoans fatty acid synthases. Residue number is based on human protein sequence.

technique enables investigation of structure and conformational dynamics of the protein and protein–drug complexes at high resolution. In particular, the use of a proteolytic cleavage site to separate *h*FASN into its sub-regions from full-length protein could serve as a platform for studying the structure of the endogenous multienzyme complex isolated from natural sources, at near atomic resolution. It may also facilitate studies of protein-protein interactions including the interaction landscape of MAT-mediated acyl modified ACP domains[30] with KR, DH, and ER domains within the modifying region of the protein. Information on how ACP recognizes *h*FASN catalytic sites during palmitate biosynthesis will be highly valuable for drug design and protein engineering efforts in both fatty acid synthase and polyketide synthase[38] family of proteins.

## Methods
### Protein expression and purification
A C-terminal 6 × His-tag containing vector harboring *h*FASN was purchased from Addgene (item # 107138) and primers for cloning were synthesized by Integrated DNA Technologies (idtdna.com). A TEV protease site was introduced into the pcDNA3.1 *h*FASN plasmid by using overlap extension PCR and confirmed by sequencing. A 500 mL HEK293F cell culture (Thermo Fisher FreeStyle™ 293-F Cells, catalog # R79007) was grown in Gibco FreeStyle medium. Cells were incubated in a New Brunswick S41i shaking incubator at 37 °C, 8% CO₂, and 125 rpm for three hours to get a total cell density of about $1.0 \times 10^{6}$ cells/mL. Cells were then transfected by mixing 250 µg pcDNA3.1 *h*FASN plasmid with 250 µl FectoPRO (PolyplusTransfection, reference 166-001, lot 12W2808FCI) in 50 mL Gibco FreeStyle medium.

The transfected cell culture was grown for 65 h using the same conditions in the New Brunswick S41i shaking incubator. After 65 h, cells were harvested by centrifuging at $3000 \times g$ for 5 min at 4 °C in a Beckman Coulter Allegra X-30R centrifuge. The supernatant was decanted, and cell pellets were collected. The cell pellets were resuspended in lysis buffer (50 mM Tris pH 7.4, 150 mM NaCl, 0.5 mM PMSF 1 mM Benzamidine, 1 mM 6-aminocaproic acid, and 0.1% (v/v) Triton X-100) and incubated for 10 min on ice with occasional agitation. The lysate was then cleared by centrifugation at $10,000 \times g$ for 15 min and the supernatant was filtered using a 0.2 µm syringe filter. The filtered supernatant was passed through Ni-NTA agarose for affinity purification. *h*FASN was eluted from the Ni-NTA agarose affinity beads using 50 mM Tris pH 7.4, 150 mM NaCl and 300 mM imidazole. The eluate was then concentrated and passed through a Superose 6 10/300 increase column. For the isolated region of the modifying *h*FASN, sequence from G855 to G2511 was synthesized harboring an N-terminal 6 × Flag-tag and C-terminal 6×His-tag in the pcDNA3.1 vector by GenScript. A 250 mL HEK293F cell culture was transfected for protein expression using a similar condition used for the full-length TEV protease hFASN construct. The transfected cell culture was harvested 48 h after post-transfection and protein purification was done using the same lysis buffer and purified using Ni-NTA agarose for affinity capture followed by gel filtration using superdex 200 increase 10/300 column.

Full length *h*FASN protein was purified to a total amount of 5 mg from a 500 ml culture with ~1 mg of total cleaved modifying region attainable after TEV cleavage and size exclusion chromatography. The truncated modifying region from the synthetic gene was purified at about 5 mg total protein from a 250 ml culture. SDS-PAGE gels of purified proteins are shown in supplementary Fig. 9.

SuperTEV protease was purified from *E. coli* BL21 cells that were transformed with a plasmid encoding 6 × His tagged superTEV[39]. *E. coli* cells were grown in media containing NZ amine A, yeast extract, and sodium chloride. TEV protease expression was induced by adding 0.1 mM IPTG. After overnight growth, the cells were harvested and lysed by sonication. The lysate was cleared by centrifugation and passed through Ni-NTA agarose. Elution fractions were pooled and diluted in storage buffer containing 50 mM Tris-HCl pH 7.5, 1 mM EDTA and 5 mM DTT. Glycerol was added up to 50% and the enzyme was stored at 1 mg/mL concentration in −80 °C.

### TEV protease cleavage
*h*FASN Superose 6 10/300 increase fractions from 14 to 16 mL elution volumes were pooled and mixed with superTEV at 1:200 (*h*FASN:TEV) molar ratio in a TEV cleavage buffer (50 mM Tris, 1 mM DTT, 1 mM EDTA). The sample was then put into a dialysis bag (Spectrum™ Spectra/Por™ 4 RC Dialysis Membrane Tubing 12,000 to 14,000 Dalton MWCO) and kept in room temperature overnight for cleavage and buffer exchange. The cleaved sample was concentrated using 100 kDa cut-off concentrators and loaded onto a Superdex 200 10/300 increase gel filtration column for separating the modifying region of *h*FASN. The quality of the purified protein was assessed using an 8–15% polyacrylamide gel.

### Activity assay
The activities of the KR domain of the *h*FASN with and without Denifanstat were determined by monitoring NADPH oxidation using a CLARIOstar spectrophotometer (software version: 5.20 R5) at 334 nm in the presence of trans-1-decalone (Sigma-Aldrich). Reaction was performed at 37 °C in 96 well microplates in a final volume of 150 µl. The assay mixture contained 50 mM sodium phosphate at pH 7.4, 1 mM DTT, 0.04% (v/v) Tween-20, 400 µM NADPH, 5 mM trans-1-decalone and 100 nM purified protein. For protein inhibition, 10 µM Denifanstat was used. The DH domain activity of the wild-type, L1097A, and H878A *h*FASN was measured spectrophotometrically through the

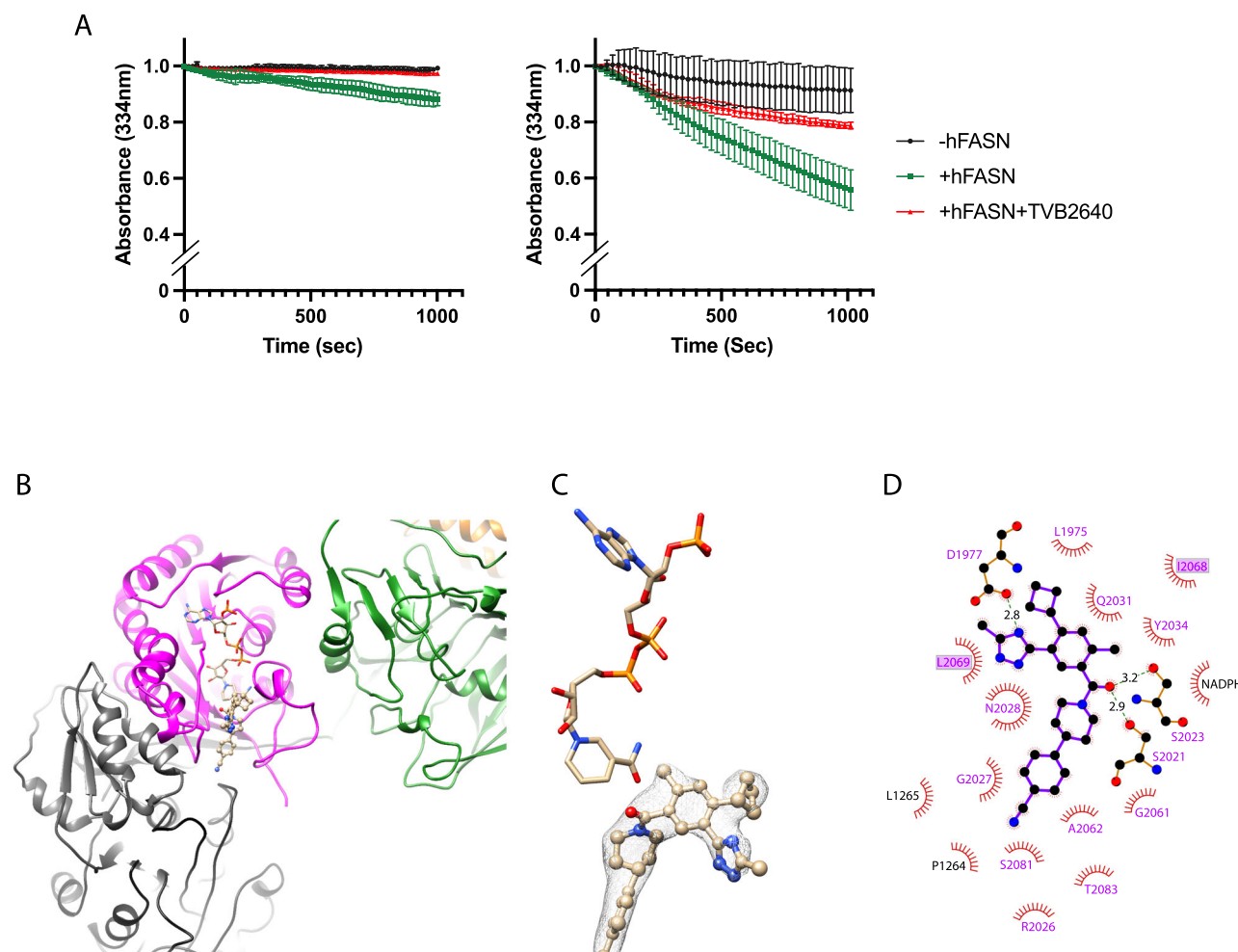

**Fig. 5 | Interaction landscape of Denifanstat. A** Inhibition of the KR activity of full-length TEV engineered *h*FASN (left) and modifying region expressed and purified as a truncated construct (right) by addition of Denifanstat. Assays are done in two biological replicates (i.e., two independent protein preparations). First 600 s are used to measure specific activity for both assays. Source data are provided as a Source Data file. **B** CryoEM model of ΨME:ΨKR:KR tridomain bound to NADPH

(shown as sticks) and Denifanstat (shown as ball and stick). **C** Model to map density for the drug carved at 2 Å (see supplementary Fig. 8 for uncarved map). **D** 2D interaction landscape for Denifanstat. Residues are colored based on domain color scheme in Fig. 1. KR residues in the predicted ACP interaction region are high-lighted within boxes. 2D interaction network is generate using LigPlot[50].

decrease in absorbance caused by NADPH oxidation at 334 nm. The reactions were carried out at 37 °C in 96 well microplates in a final volume of 150 μl. The reactions mixture contained 50 mM sodium phosphate at pH 7.4, 1 mM TCEP, 1 mM EDTA, 40 μM NADPH and 800 μM DL-beta-Hydroxybutyryl CoA (Sigma-Aldrich) and 100 nM purified protein. A calibration curve was taken for 5 min before adding the substrates. The error of the graph was plotted from the two bio-logical replications with technical duplicate conditions in each purified protein preparation unless otherwise stated. More specifically, two protein preparations from two separate transfections were used (i.e., biological duplicate). For each biological replicate, two independent assays were run on the same 96-well plate (i.e., technical duplicate). Each curve is normalized against its first point of measurement after the addition of the substrates. Each assay replicate is plotted indivi-dually in Supplementary Fig. 10. All assay data are provided as excel files containing raw and normalized absorbance values as well as cal-culated NADPH concentrations at each time point.

### Cryo-EM sample preparation and image collection
For electron cryo-microscopy of the NADPH bound *h*FASN modifying region prepared after TEV protease cleavage, 3 μl of 0.8 mg/mL *h*FASN modifying region, 1 mM NADPH, and 1 mM DTT was applied

onto in-house nanofabricated holy gold grids[40]. For cryoEM of the inhibitor bound complex *h*FASN modifying region was incubated with 13.75 μM Denifanstat, 1 mM NADPH and 1 mM DTT for 1 h before freezing. For the isolated modifying region of *h*FASN that was inde-pendently transfected and purified, the inhibitor bound complex was made by adding 1 mM NADPH, 1 mM TCEP and 22.2 μM Denifanstat. All freezing was done using the plunge method into liquid ethane held at liquid nitrogen temperatures with a Vitrobot Mark IV (FEI) at 3.5 s blotting, a blot force of +3, 4 °C, and 100% relative humidity. Cryo-grids were screened on a ThermoFisher (TFS) Talos L120C equipped with a LaB₆ emitter and a Ceta CCD camera. Samples demonstrating optimal vitreous ice quality and particle distribution were used for data collection at a TFS Titan Krios electron microscope using EPU software version 3.3.1.5184REL (Supplementary Table 1).

### Image processing and 3D reconstruction
Image processing were performed on cryoSPARC V3[41]. Electron micrograph movies were aligned and averaged using patch motion correction. Contrast transfer function (CTF) was estimated using the patch CTF algorithm. Particle box size was set at 250 pixels (1.03 Å/pixel). Manual particle picking was challenging due to the elongated

nature of the complex that resulted in different projection images depending on the viewing direction. Therefore, particle picking was done via template-based picking, where templates were generated using simulated particle images of *S. scrofa* FASN core modifying region (PDB: 2vz9)[4] low pass filtered to 20 Å. Contaminants and broken particle images were cleared via 2D classification and 3D ab initio reconstructions for NADPH bound and NADPH + TVB2640 bound *h*FASN samples, respectively.

An initial 3D consensus refinement was performed to assign orientation parameters to the selected particle images followed by 3D classification without orientation search to further clean the particle data set and identify 3D reconstructions with optimal viewing direction. Resolution anisotropy was qualitatively assessed by observing side chain densities from different directions (Figs. 3 and S6D). Map sharpening was done using sharpening tool in cryoSPARC V3 and setting the B-factor to −50. B-factors determined by Guinier plot analysis was at −119.6 and −120.3 for NADPH and NADPH + TVB2640 samples, respectively. B-factor of −50 was used to reduce the noise in the sharpened maps. 3DVA was done on selected high quality 3D reconstructions using default parameters (i.e., number of modes to solve = 3), except for limiting the resolution at which results were filtered to 4 Å. 3DVA results were then displayed using linear mode in cryoSPARC V3.

## Model building

A homology model for the core modifying region of human fatty acid synthase was generated using SWISS-MODEL using *S. scrofa* FASN (PDB: 2vz9)[4] as the template. This homology model was flexibly fit into the cryoEM density sharpened with a B-factor of −50 for each of the complexes of *h*FASN modifying region, except for the C2 refined map (Supplementary Table 1) where a B-factor value determined via Guinier plot analysis was used. Flexible fitting was done with real space refinement in Phenix v1.20.44.59[42] and the model to map fit was assessed and manually modified in Coot v0.9.6[43] in an iterative process. Amino acids missing side chain densities were modeled as stubs (only beta-carbon is shown, rest of side chain is omitted), with their amino acid identity maintained. Refinement statistics can be found in Supplementary Table 1. Flexible fitting into the cryoEM maps generated via 3DVA analysis was done using iMODFIT[44] implemented in UCSF Chimera v1.16[45]. Model and map visualization was done in PyMol v4.6.0[46] and UCSF Chimera v1.16[45]. Cavities were visualized in PyMol using a cavity detection radius and cut-off of 7 Å and 3 solvent radii, respectively.

## Sequence alignment

250 Nonredundant amino acid sequences from Metazoa (taxid:33208) excluding *Homo sapians* (taxid:9606) were aligned against full *h*FASN using BLASTp[47]. Multiple sequence alignment was done using Clustal Omega[48] and LOGO plot was generated via the WebLogo server[49]. Y-axis of the LOGO plot was set to represent frequency of amino acids plotted against amino acid number (based on *h*FASN sequence) on the x-axis.

## Data availability

All data and materials are available upon request. CryoEM density maps and atomic models have been deposited in the Electron Microscopy Data Bank and the Protein Data Bank under accession codes: EMD-28690, EMD-28691, EMD-40182, 8EYI, 8EYK, and 8GKC. The NADPH oxidation data generated in this study are provided in the Source Data file. Source data are provided with this paper.

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

## Acknowledgements

This work was supported by Canadian Institutes of Health Research (CIHR) project grant number: 419240, a discovery grant from the Natural Sciences and Engineering Research Council of Canada (NSERC, RGPIN-2018-06070), and Princess Margaret Cancer Foundation (PMCF). S.M.N.H. was supported by an Ontario Graduate Scholarship award and PMCF. J.W.L. and D.L.D. were supported by PMCF and NSERC CGS-D awards. M.T.M.J. and A.K. were supported by PMCF. We thank the Toronto High Resolution High Throughput cryo-EM facility, supported by the Canada Foundation for Innovation and Ontario Research Fund, for cryo-EM data collection. The Talos L120C was funded by Canada Foundation for Innovation (CFI) and Ontario Research Fund-Research Innovation (ORF-RI).

## Author contributions

S.M.N.H.: study design, cloning, protein purification, cryoEM, data analysis, activity assays, manuscript writing. J.W.L.: cloning, protein purification, A.K.: cryoEM, data analysis. D.L.D.: manuscript editing. M.T.M.J.: study design, data analysis, manuscript writing.

## Competing interests

The authors declare no competing interest.
