## [Peer Review File · Nature Communications]

REVIEWER COMMENTS

Reviewer #1 (Remarks to the Author):

Fatty acid biosynthesis is a key metabolic pathway in eukaryotes and prokaryotes. While prokaryotes perform synthesis with separate enzymes, eukaryotic cytoplasmatic fatty acid synthesis is achieved within multienzymes. Methodological progress in X-ray crystallography and more recently in Cryo-EM has spurred the structural characterization of the fatty acid synthase (FAS) multienzyme of eukaryotes. Fungal FAS has been characterized in atomic detail by several landmark papers, from Jenni et al. (2006, *Science*) to Singh et al. (2020, *Cell*), and also mammalian FAS has been analyzed in structural details from 2006 onwards, with most important contributions by the labs of Ban and Maier by crystal structures on the FAS from pig (Maier et al. 2006 and 2010, both *Science*). The human FAS is a relevant target in therapeutic treatment, and domains and didomains have been solved at high resolution; i.e., the KS-AT subunit by Pappenberger et al. and the ME-KR subregion by Hardwicke et al. For improving the understanding of human fatty acid biosynthesis and for informing drug design, the full-length structure of the core catalytic domains of FASs has always been considered a relevant structure biological target, however, the successful structure solution remained unmatched until now.

Mazhab-Jafari and coworkers have already proved expertise in the structural characterization of FASs, by giving insight into the substrate shuttling in the fungal megasynthase. Now, they present a breakthrough in the structural characterization of mammalian FAS, and present for the first time human FAS in full length (in two parts). The separation of the FAS in the condensing and processing wing by proteolytic cleavage does not limit the relevance of the structure significantly, since it is known that the two wings are structurally autarkic. They are flexibly connected and interact just via transient docking of the acyl carrier protein (ACP). In this regard, the work of Mazhab-Jafari and coworkers is surely a landmark contribution to the structural understanding of human FAS.

The strength of the work lies in the approach to access near-atomic resolution data of human FAS. It involves the *E. coli* recombinant expression of a C-terminally His-tagged version of the human FAS, which in addition harbors a TEV cleavage site between the condensing and the processing wing. The purified protein is then treated with TEV protease at high molar ratio, and separated in wings by size-exclusion chromatography.

The structural core of the processing wing (modifying region) of human FAS was solved to 2.7 Å resolution. Mazhab-Jafari and coworkers then focused on the domains dehydratase (DH) and the ketoreductase (KR) in two aspects; first, in revealing a different binding channel in DH in human FAS than observed in porcine FAS, and in presenting KR in complex with the anti-cancer drug Denifanstat. Indeed, demonstration of this application moves this paper from a pure basic research study to a higher impact work of translational relevance. I would recommend the publication of this manuscript after the following points are fixed in a revision:

(1) The following papers should be acknowledged and included as reference:

(i) Line 84: The ACP domain of the human FAS has been solved (in complex with PPT);
<https://doi.org/10.1016/j.chembiol.2007.10.013>

(ii) Line 107: It has been shown for the highly homologous FAS from mouse that the N-terminal condensing wing is not required for the recombinant expression of the modifying core of the processing wing. The protein yield is high and SDS-PAGE demonstrates proteolytic stability.
<https://doi.org/10.1002/pro.3550>

(iii) It is recommended to also include, the probably most recent review about fatty acid biosynthesis, which is well suited for the broad readership of *Nat Commun*;
<https://pubs.acs.org/doi/10.1021/acs.chemrev.1c00147>

(2) The Fig. S1 shows a relatively large shift in apparent molecular weight upon insertion of the TEV cleavage site. How can this be explained on the basis of protein properties? On the other hand, the shift may just result from a slightly different instrument set-up (different column, different tubing, ...). Indeed, the synchronous shift of an aggregate peak suggests technical reasons. Ideally calibrations for both runs are included. Can the authors clarify this point?

(3) Is there any specific reason why the human FAS was expressed without PPT (e.g. Sfp as used from mouse FAS in Rittner et al. Protein Science, 2019) leading to fatty-acid-inactive non-phosphopantetheinylated protein? Active protein would enable superior protein quality checks, as well as monitoring effects of inhibitors. Thinking the presented approach as platform for structural and, ideally, also functional and inhibitory studies, active protein would be important.

(4) Figure 5 A: The KR activity assays is performed at high absorbance, probably outside the linear range. Is there any specific reason for this assay set-up?

(Note) Movies were not provided for review, such that the informed assessment of the conformational variability was not possible. Cryo-EM data have been checked, but this referee is not an EM expert.

Reviewer #2 (Remarks to the Author):

In their manuscript "Atomic model for core modifying region of human fatty acid synthase in complex with Denifanstat" Naimul Hassan and coauthors report cryoEM reconstructions of a proteolytically derived large fragment of the human fatty acid synthase, both in complex with only the cofactor NADP and with NADP and the drug candidate denifanstat.

The manuscript is most strongly focused on reporting two highly similar cryoEM reconstructions with and without a drug candidate bound to the KR active site. It draws its relevance (1) from reporting a system for the production of the fragment by introducing a TEV protease cleavage site into a linker peptide region and (2) enabling "atomic-resolution structure determination of small molecule drugs complexed with the core modifying region (of FAS). Given the close structural similarity to earlier models, the protein structure itself appears as less exciting. The relevance of a minor structural difference by a point mutation around the DH active site remains unclear, and is not functionally explored in any way.

For (1) the authors don't report any own or quantitative data on comparison to other options for sample production, in particular the simple approach of expressing the "modifying region" directly in a similar system. This has been successful in related systems, despite the early reference quoting problems with such constructs from times before the overall structure of FAS was known. While their approach was successful for cryoEM, where smallest quantities are analyzed with in silico purification, it appears as highly inefficient. The sample, based on SDS-PAGE was of modest purity with many bands, and the protease cleavage was highly inefficient, requiring a 1:200 ratio of target to protease. Such process appears rather unfavorable for rational drug development, as the resulting sample may not qualify for other means of structure determination or biophysical assays at larger scale. At least, the authors should provide yield and purity estimates, and compare those estimates as well as the observed catalytic efficiency to reported literature values.

Regarding (2), The authors focus on small molecule visualization at high resolution, described correctly as "providing an atomic model for ...", but incorrectly in the text as "atomic resolution". Even the overall resolution of around 2.7Å is not atomic resolution, even though an atomic model can be built and refined based on restraints even at this lower resolution. The visualization of the ligand looks like slightly worse than 3Å local resolution, with smaller substituents hard to delineate, and is of borderline quality for structure-based drug design. The authors should avoid overstating map quality by describing the results as atomic resolution, and should provide the local resolution in the ligand region. Overall, the cryoEM reconstructions and model refinement use standard protocols and tools.

Further comments:

L24: Conservation of fatty acid biosynthesis in all kingdoms of life: Authors might tone down for archaea, see e.g. here: <https://doi.org/10.1111/1462-2920.12359>

L27: individual catalytic domains. should be: individual catalytic proteins

L30: multi-domain complexes: Complexes of multi-domain protomers ?

L31: what is exactly meant by repeating ?

L35: Fig. 1A check graphics quality (ACP at KR)

L68: Linker between KS and DH. Note that the linker in sequence space is between the (M)AT and DH domain, in related PKS it is often referred to as the post-AT linker. Even though the linker in the 3-dimensional structure is located between KS and DH, the term KS-DH linker may be misleading.

L89: "has not been solved to atomic resolution": that is correct, although an apparently accurate atomic model has been provided before. Note, that also here no atomic resolution is obtained, and the resolution is only moderately improved (3.2A -> 2.7 A), while for the purpose of drug development the ligand-binding KR domain has already been visualized in a different construct at 2.3A resolution.

L93: "enables cleavage": cleavage appears to be very inefficient, this should be noted even in such summary statement.

L121: Fig 2A: Have contaminant bands been analyzed to determine identity as either degradation products of FAS or independent contaminations ?

L121: Fig 2C: The amount of useful "roughly 1:1 cleavage product of modifying and condensing region" in only one fraction indicates highly inefficient cleavage and low yields despite excessive amounts of protease added. Please quantify yields at relevant steps of protein production.

L121: Fig 2D: 2D classes look poor compared to those shown in supplement, probably from later stage 2D classes ?

L152: Do the authors see any functional relevance to the structural difference resulting from a leucine to alanine substitution ? This could be tested by mutational analysis.

L153: Fig 4: The authors indicate "entry" and "exit" with arrows for a substrate in Pig DH. How do the authors depicted such motion of the substrate that during catalysis at this active site is covalently tethered to the bulky ACP Ppant cofactor through the narrow tunnel ?

L163: Have the authors also considered to run 3DVA for more than 3 modes ?

L198: Activity of KR: Compare to literature values.

L206: "was generate at" : language

L214: Fig 4A: The assay shows plots with unusual and difficult to reconcile jumps in the curve. The selection of the linear range for determining the initial velocity may be affected by this. Overall the Absorption is very high, (2.2-2.6) indicating suboptimal assay conditions.

L214, Fig. 4C: Carved map: Appearances is of around 3.2-3.5A local resolution, certainly not atomic resolution.

L247: provide yields, see above.

L262: 8-15% agarose gel : SDS polyacrylamide gel ?

L270: replicate: text is not fully clear to me, is this what is meant: Purified and processed samples from two different transfections were used, and for each of those to independent assays were pipetted on different plates ?

We thank both reviewers for their valuable critique that has helped us improve the manuscript considerably. Please see our point-by-point response below. Reviewer comments are *italicized and underlined*. All changes in the revised manuscript are tracked in the word file.

Reviewer #1 (Remarks to the Author):

Fatty acid biosynthesis is a key metabolic pathway in eukaryotes and prokaryotes. While prokaryotes perform synthesis with separate enzymes, eukaryotic cytoplasmatic fatty acid synthesis is achieved within multienzymes. Methodological progress in X-ray crystallography and more recently in Cryo-EM has spurred the structural characterization of the fatty acid synthase (FAS) multienzyme of eukaryotes. Fungal FAS has been characterized in atomic detail by several landmark papers, from Jenni et al. (2006, Science) to Singh et al. (2020, Cell), and also mammalian FAS has been analyzed in structural details from 2006 onwards, with most important contributions by the labs of Ban and Maier by crystal structures on the FAS from pig (Maier et al. 2006 and 2010, both Science). The human FAS is a relevant target in therapeutic treatment, and domains and didomains have been solved at high resolution; i.e., the KS-AT subunit by Pappenberger et al. and the ME-KR subregion by Hardwicke et al. For improving the understanding of human fatty acid biosynthesis and for informing drug design, the full-length structure of the core catalytic domains of FASs has always been considered a relevant structure biological target, however, the successful structure solution remained unmatched until now.

Mazhab-Jafari and coworkers have already proved expertise in the structural characterization of FASs, by giving insight into the substrate shuttling in the fungal megasynthase. Now, they present a breakthrough in the structural characterization of mammalian FAS, and present for the first time human FAS in full length (in two parts). The separation of the FAS in the condensing and processing wing by proteolytic cleavage does not limit the relevance of the structure significantly, since it is known that the two wings are structurally autarkic. They are flexibly connected and interact just via transient docking of the acyl carrier protein (ACP). In this regard, the work of Mazhab-Jafari and coworkers is surely a landmark contribution to the structural understanding of human FAS.

The strength of the work lies in the approach to access near-atomic resolution data of human FAS. It involves the E. coli recombinant expression of a C-terminally His-tagged version of the human FAS, which in addition harbors a TEV cleavage site between the condensing and the processing wing. The purified protein is then treated with TEV protease at high molar ratio, and separated in wings by size-exclusion chromatography.

The structural core of the processing wing (modifying region) of human FAS was solved to 2.7 Å resolution. Mazhab-Jafari and coworkers then focused on the domains dehydratase (DH) and the ketoreductase (KR) in two aspects; first, in revealing a different binding channel in DH in human FAS than observed in porcine FAS, and in presenting KR in complex with the anti-cancer drug Denifanstat. Indeed, demonstration of this application moves this paper from a pure basic research study to a higher impact work of translational relevance. I would recommend the publication of this

manuscript after the following points are fixed in a revision:

We thank the reviewer for positive feedback and assessment of the manuscript and recommendation for publication. We also thank the reviewer for the critique and comments provided below. All recombinant proteins in this study were over-expressed and purified from an engineered non-adherent human embryonic kidney cell line (i.e., HEK293F). Please see our point-by-point response below.

(1) The following papers should be acknowledged and included as reference:

(i) Line 84: The ACP domain of the human FAS has been solved (in complex with PPT);

<https://doi.org/10.1016/j.chembiol.2007.10.013>

Yes, the reviewer is correct, and we have missed this reference in the initial manuscript version. It is now added to the revised manuscript (line 86).

(ii) Line 107: It has been shown for the highly homologous FAS from mouse that the N-terminal condensing wing is not required for the recombinant expression of the modifying core of the processing wing. The protein yield is high and SDS-PAGE demonstrates proteolytic stability.

<https://doi.org/10.1002/pro.3550>

We thank the reviewer for this reference. Please see our response to reviewer 2 regarding this point. Indeed, we were aware of this observation from this paper but did not include it in the first draft since we could not find any data on dimerization status of the construct. We have now included this reference and mentioned the proteolytic stability of mouse FASN processing wing (i.e., modifying region) observed in the SDS-PAGE, which suggests proper dimer formation (line 112-114).

(iii) It is recommended to also include, the probably most recent review about fatty acid biosynthesis, which is well suited for the broad readership of Nat

Commun; <https://pubs.acs.org/doi/10.1021/acs.chemrev.1c00147>

Agreed. This is an excellent review. It is now added in the revised manuscript (line 34).

(2) The Fig. S1 shows are relatively large shift in apparent molecular weight upon insertion of the TEV cleavage site. How can this be explained on the basis of protein properties? On the other hand, the shift may just result from a slightly different instrument set-up (different column, different tubing, ...). Indeed, the synchronous shift of an aggregate peak suggests technical reasons. Ideally calibrations for both runs are included. Can the authors clarify this point?

The instrument set up between the two runs were identical. The shift in the apparent molecular weight upon insertion of the TEV cleavage site can be due to increase flexibility of the engineered construct, since the length of the unstructured loop between the condensing and modifying

regions will increase by seven amino acids, which is the recognition site for TEV protease. We also noticed the size exclusion column was labeled incorrectly in the figure caption. It is now corrected to Superose 6 increase 10/300.

(3) Is there any specific reason why the human FAS was expressed without PPT (e.g. Sfp as used from mouse FAS in Rittner et al. Protein Science, 2019) leading to fatty-acid-unactive non-phosphopantetheinylated protein? Active protein would enable superior protein quality checks, as well as monitoring effects of inhibitors. Thinking the presented approach as platform for structural and, ideally, also functional and inhibitory studies, active protein would be important.

This is a valid point, and we will certainly perform co-expression studies with PPT proteins such as Sfp in our future studies. The focus of this study was to demonstrate the ability of high-resolution structure determination of the core modifying region of human fatty acid synthase and assessment of its conformation dynamics and interaction with small molecule inhibitor, Denifanstat. We agree with the reviewer that expression of activated hFASN will improve protein quality checks and it may further improve yield and purity as well. It is worth noting that hFASN preparations from HEK293F cells are at least partially phosphopantetheinated as we detect enzymatic-dependent NADPH oxidation when the TEV engineered full length protein is supplied with 3-hydroxybutyryl-CoA and NADPH. See our reply to second reviewer below and revised supplementary figure 4.

(4) Figure 5 A: The KR activity assays is performed at high absorbance, probably outside the linear range. Is there any specific reason for this assay set-up?

We agree with the reviewer that the reported absorbance values were high in the initial manuscript. However, we would like to note that the activity assays were done on a plate reader with "path length correction" turned On, which normalizes the absorbance values from the well to those from a standard cuvette. Our measurements without pathlength correction are at ~1.0 at the start of the assay at 334 nm. Please see the following link for more information regarding path length correction (<https://tools.thermofisher.com/content/sfs/brochures/AN-SkanIT-Microplate-Based-Pathlength-Correction-Technical-Note-EN.pdf>).

(Note) Movies were not provided for review, such that the informed assessment of the conformational variability was not possible. Cryo-EM data have been checked, but this referee is not an EM expert.

Movies from the conformational variability analysis are embedded as PowerPoint files in the manuscript word file. They are accessible by double clicking movie images in the word file (lines 233 and 238 for the main manuscript and lines 103, 107, and 112 in the supplementary material). Additionally, in the revised submission, we have submitted separate PowerPoint files for each movie for the ease of access.

Reviewer #2 (Remarks to the Author):

In their manuscript "Atomic model for core modifying region of human fatty acid synthase in complex with Denifanstat" Naimul Hassan and coauthors report cryoEM reconstructions of a proteolytically derived large fragment of the human fatty acid synthase, both in complex with only the cofactor NADP and with NADP and the drug candidate denifanstat.

The manuscript is most strongly focused on reporting two highly similar cryoEM reconstructions with and without a drug candidate bound to the KR active site. It draws its relevance (1) from reporting a system for the production of the fragment by introducing a TEV protease cleavage site into a linker peptide region and (2) enabling "atomic-resolution structure determination of small molecule drugs complexed with the core modifying region (of FAS). Given the close structural similarity to earlier models, the protein structure itself appears as less exciting. The relevance of a minor structural difference by a point mutation around the DH active site remains unclear, and is not functionally explored in any way.

Thank you to the reviewer for insightful critique of the manuscript. We reported the difference between the DH cavities of human and porcine FASN proteins as a natural structural variation in metazoan type I FASN. This has now been clarified in the revised manuscript (lines 212-216). Therefore, we anticipate that point mutations of human FASN at L1097 residue to those amino acids found in other species may not inhibit the DH enzymatic function. To address the reviewer's request, we have analyzed the human FASN L1097A mutant by functional assay. This mutation replaces the leucine residue with an alanine found in porcine FASN.

To measure DH activity, we incubated TEV engineered FASN wild type, L1097A, and H878A (DH inactive mutant), with 800 μ M of 3-hydroxybutyryl-CoA and 40 μ M of NADPH. Full length TEV engineered proteins were used for this assay (i.e., no TEV cleavage). We then measured NADPH oxidation by ER domain at 334 nm. This was therefore a DH and ER coupled assay as described before (PMID: 9047334). MAT domain transacylates ACP domain using 3-hydroxybutyryl-CoA with approximately similar efficiency as to those of acetyl- and malonyl-CoA (PMID: 29328619). Hydroxybutyryl is then converted to crotonyl by DH and subsequently reduced by the NADPH-dependent reduction reaction catalyzed by ER. We confirmed that NADPH oxidation was dependent on catalytic activity of DH domain as FAS H878A mutant showed no reduction in absorbance at 334 nm (revised supplementary Fig 4A). We also showed that this assay was dependent on the presence of MAT domain as there was no observable NADPH oxidation in the presence of the isolated FASN modifying region (DH-ER-KR-ACP-TE) (revised supplementary fig 4B). There was no significant change in NADPH oxidation upon L1097A mutation. Considering similar specific activities of DH and ER domains of human FASN (PMID: 16215233), this assay indicates that L1097A mutation does not significantly impact DH catalytic function. We have added supplemental figure 4 and discuss these observations in the revised manuscript (lines 212-216).

For (1) the authors don't report any own or quantitative data on comparison to other options for

sample production, in particular the simple approach of expressing the “modifying region” directly in a similar system. This has been successful in related systems, despite the early reference quoting problems with such constructs from times before the overall structure of FAS was known. While their approach was successful for cryoEM, where smallest quantities are analyzed with in silico purification, it appears as highly inefficient. The sample, based on SDS-PAGE was of modest purity with many bands, and the protease cleavage was highly inefficient, requiring a 1:200 ratio of target to protease. Such process appears rather unfavorable for rational drug development, as the resulting sample may not qualify for other means of structure determination or biophysical assays at larger scale. At least, the authors should provide yield and purity estimates, and compare those estimates as well as the observed catalytic efficiency to reported literature values.

We thank the reviewer for raising the possibility of alternate sample preparation for the modifying region of FASN. In response to the reviewer suggestion, we tested expression and purification of the modifying region alone in the same expression system. The protein was successfully expressed and purified (revised Figure 2D) with a yield ten times higher than that of the TEV cleavage method described in the initial version of the manuscript (i.e., method 1). Additionally, the protein was significantly purer (>95%) than the method 1 preparation as judged by SDS-PAGE followed by Coomassie staining. Therefore, expression and purification of isolated modifying region of human FASN (i.e., method 2) is more ideal for downstream drug screening and development applications compared to method 1. We also tested method 2 preparation for KR catalytic activity using trans-1-decalone as substrate and calculated specific activity (line 253). Additionally, we characterized the FASN modifying region from method 2 preparation by cryoEM and found that the protein maintains native global fold (revised figure 2D). As yield & purity is an important factor for drug screening, we include this insight in our paper and point out its benefits/drawbacks versus our approach, the main points being:

A- Expression of truncated modifying region of FASN alone in human cells may alter cellular signaling network and metabolism compared to when full length protein is expressed. This was evident in our revised work here as HEK293F cells expressing the truncated modifying region (method 2) were generally much healthier (i.e., 2.3 million cells/ml at 85% viability) than those expressing full length FASN (either WT or the TEV inserted constructs with 1.2 million cells/ml at 55% viability) at the time of harvest as judged via a dye exclusion test using Trypan blue solution. Minimal perturbation to metabolism is especially important if a researcher is interested to structurally study naturally occurring FASN post-translational modifications such as Lactylation (PMID: 36651176), acetylation (PMID: 36459649), phosphorylation (PMID: 21080941), S-nitrosylation (PMID: 26851298), O-GlcNAcylation (PMID: 27185461), and ubiquitination (PMID: 23269672) in a healthy or a cancer cell. It is also important if the goal is to structurally characterize native intermolecular interaction of FASN under physiological conditions. Endogenous FASN can be purified to ~90% purity from cancer cells as was shown by Wah Chiu's group in 2002 (PMID: 11756679). We have recapitulated these results using similar protocols as described by Chiu's group for a luminal-A type adherent breast cancer cell line (see figure below) where we can purify 1 mg of endogenous FASN (i.e., no affinity tags) from fifty 182 cm² culture

plates and analyse the full-length protein using cryoEM. We have identified an endogenous interacting protein with FASN from this cancer cell line (we would like not to reveal the protein identity for confidentiality). One can then use CRISPR to generate cell lines expressing TEV engineered FASN constructs harbouring either N- or C-terminal affinity tags expressed from native FASN promoter and terminator. Cleavage of the condensing and modifying regions should enable high-resolution cryoEM study of these endogenous proteins under physiological or diseased conditions. cryoEM is a versatile structure determination technique for scarce and endogenous proteins that can only be purified to low abundance and purity. We have discussed this point in the revised manuscript (lines 145-153 and 154-172).

Figure caption: from left to right: SDS-PAGE followed by Coomassie stain analysis of endogenous FASN purified from a luminal-A breast cancer cell line by ion exchange followed by size exclusion chromatography of cytosolic cell fraction. Representative cryo electron micrograph of endogenous FASN particles (red circles) with a representative 2D class and a 3D ab initio reconstruction at 25 Å resolution. The small interacting protein is not distinguishable at this resolution. Data collected on a 200kV screening electron microscope.

B- Investigating transient and low populated states of recombinant FASN modifying region with MAT-mediated acyl-modified ACP domains. More specifically, one can utilize the substrate polyspecificity of the MAT domain (PMID: 29328619) that is in the condensing region of FASN to acylate the holo ACP domain residing in the modifying region with unusual CoA esters (i.e., mimicking FASN reaction intermediates). Holo form of ACP can be generated by the action of a PPTase such as Sfp or AcpS. PPTases generally have lower K_{cat}/K_m for the phosphopantetheination reaction of apo ACP domains with coenzyme A (PMID: 24292120) compared to the K_{cat}/K_m of MAT domain acylation reaction of holo ACP domain using acyl-CoAs (PMID: 29328619). Following the MAT-acylation reaction, the condensing and modifying regions can be proteolytically separated to enable cryoEM based investigation of transient interactions between the acyl modified ACP with catalytic domains in the modifying regions that are inactivated by active site mutations (i.e., KR, DH, and ER). cryoEM is a versatile method to study transient interactions given a large dataset of particle images (e.g., 5×10^6 for FASN, which can be collected in two to three days at a high-end electron microscope operating at 300kV). We have demonstrated the ability to construct 3D cryoEM density maps of stalled ACP domain in fungal FAS (PMID: 32471977) using a small dataset (10^5 particle images) collected using a medium-end electron microscope operating

at 200kV. Given the TEV cleavage method of protein preparation, one can then acylate ACP domain using a variety of CoA esters, all in one complex (i.e., full length holo FASN) followed by proteolytic cleavage and cryoEM studies of ACP interactions with inactivated KR, DH, and ER domains. We have discussed this point in the revised manuscript (lines 172-183).

Regarding (2), The authors focus on small molecule visualization at high resolution, described correctly as “providing an atomic model for ...”, but incorrectly in the text as “atomic resolution”. Even the overall resolution of around 2.7Å is not atomic resolution, even though an atomic model can be built and refined based on restraints even at this lower resolution. The visualization of the ligand looks like slightly worse than 3Å local resolution, with smaller substituents hard to delineate, and is of borderline quality for structure-based drug design. The authors should avoid overstating map quality by describing the results as atomic resolution, and should provide the local resolution in the ligand region. Overall, the cryoEM reconstructions and model refinement use standard protocols and tools.

The reviewer is correct regarding the description of “atomic resolution”. We have therefore replaced this phrase with either “near atomic-resolution” or “high-resolution” throughout the revised manuscript. In regard to the quality of the cryoEM map in the Denifanstat drug binding site within the KR domain, it is important to note that all the maps shown in the manuscript are sharpened using a B-factor that is more positive than the ones calculated for each map via Guinier plot analysis (PMID: 14568533) that is implemented in cryoSPARC software (i.e., less sharpening). This was mentioned in the method section of the manuscript’s first draft (lines 381-384, revised manuscript numbering), and it was done to homogenize the sharpening factor between the different cryoEM map reconstructions and reduce noise.

To address the reviewer’s concern regarding the quality of the cryoEM density in the drug binding site we have taken advantage of the C2 symmetry of the FASN modifying region. We hypothesize that the KR, DH, and ER domains within this region of the complex should benefit from C2 symmetry since they showed minimal structural perturbations in the 3D variability analysis. CryoEM map refinement with imposing C2 symmetry marginally improved the global resolution from 2.63 Å to 2.45 Å (revised figure S7). Model to density fit is shown below for the Denifanstat drug binding site in the KR domain for both the C1 and C2 cryoEM maps, both sharpened based on their calculated B-factor via Guinier plot analysis. We also show the local resolution estimate for each map color coded onto the non-sharpened half maps. We have submitted the C2 refined map of FASN in the presence of NADPH and Denifanstat to Protein Data Bank (PDB code: 8GKC), modified supplementary table 1 accordingly, and discussed these results in the revised manuscript (lines 264-266). We have also updated figure 5C and supplementary figure 8 to include density from the C2 refined map.

Figure caption: Comparing cryoEM density maps of the KR ligand binding site with A) C1 and B) C2 refinements. Local resolution estimates are color coded on non-sharpened half-maps and model to map fit uses cryoEM maps sharpened by B-factors determined from Guinier plot analysis. Denifanstat is shown by ball and stick.

Further comments:

L24: Conservation of fatty acid biosynthesis in all kingdoms of life: Authors might tone down for archaea, see e.g. here: <https://doi.org/10.1111/1462-2920.12359>

We thank the reviewer for sharing this study with us. We have modified the statement in the revised manuscript as follow: "De novo fatty acid synthesis is a conserved multi-step reaction (Fig 1A) found in majority of prokaryotic and eukaryotic species".

L27: individual catalytic domains. should be: individual catalytic proteins

Agreed and modified in the revised text.

L30: multi-domain complexes: Complexes of multi-domain protomers ?

Yes we agree with multi-domain protomers and have modified in the revised text.

L31: what is exactly meant by repeating ?

Here the word 'repeating' refers to copies of a polypeptide. For example, in metazoan FASN, two copies of one polypeptide forms the complex and in most fungal species six copies of each of two polypeptides (twelve in total) form the fungal FAS complex.

L35: Fig. 1A check graphics quality (ACP at KR)

We have modified the ACP graphics at the KR domain in the revised figure 1A.

L68: Linker between KS and DH. Note that the linker in sequence space is between the (M)AT and DH domain, in related PKS it is often referred to as the post-AT linker. Even though the linker in the 3-dimensional structure is located between KS and DH, the term KS-DH linker may be misleading.

We thank the reviewer for pointing out this observation. We have modified the statement in the revised text to indicate that the linker is between MAT and DH domain.

L89: "has not been solved to atomic resolution": that is correct, although an apparently accurate atomic model has been provided before. Note, that also here no atomic resolution is obtained, and the resolution is only moderately improved (3.2A -> 2.7 A), while for the purpose of drug development the ligand-binding KR domain has already been visualized in a different construct at 2.3A resolution.

Yes, we agree with the reviewer and have cited the study that simulated the structure of human FASN (reference 13) and the first study that solved the crystal structure of a ligand-bound state of the truncated KR construct of human FASN (reference 19).

L93: "enables cleavage": cleavage appears to be very inefficient, this should be noted even in such summary statement.

We have added the statement: "Excess molar ratio of TEV relative to FASN is required to achieve ~90 % cleavage." (lines 96-97) in the revised manuscript.

L121: Fig 2A: Have contaminant bands been analyzed to determine identity as either degradation products of FAS or independent contaminations ?

In response to the reviewer's request, we have performed the following Western blot analysis using antibodies that are specific to the N-, middle, and C-terminal sequences of the recombinant protein. Please see below the results. Generally, degradation is low and can be seen at long exposures of the blot membranes (e.g., Antibodies against the N-terminal FASN sequences). Therefore, majority of bands are independent contaminants as can be seen from Coomassie staining and there is limited FASN related protein degradation.

Figure caption: Analysis of purified full-length TEV-engineered FASN protein prior to TEV cleavage. White pixels in the two left blots indicate over-exposure. Epitopes recognized by the primary antibody in each blot is indicated on the top. Residue numbers are based on sequence of human FASN protein. From left to right: primary antibody used are F1804 anti-FLAG, AF5927 anti-FASN, CST#3189 anti-FASN, CST2365T anti 6×His, and AB99359 anti-FASN.

L121: Fig 2C: The amount of useful "roughly 1:1 cleavage product of modifying and condensing region" in only one fraction indicates highly inefficient cleavage and low yields despite excessive amounts of protease added. Please quantify yields at relevant steps of protein production.

In response to the reviewer's request, we have now indicated the total amount of full length FASN purified from HEK293F cells. We have also indicated the amount of FASN modifying region that is purified after incubation with TEV protease followed by size exclusion chromatography. This information is indicated in revised experimental section (lines 320 to 323).

L121: Fig 2D: 2D classes look poor compared to those shown in supplement, probably form later stage 2D classes ?

Yes, the reviewer is correct. 2D classes shown in figure 1 of the original manuscript were from a Talos L120C screening electron microscope. We have replaced them with selected high quality 2D classes from Titan Krios electron microscope.

L152: Do the authors see any functional relevance to the structural difference resulting from a leucine to alanine substitution ? This could be tested by mutational analysis.

As discussed earlier, we tested the effect of L1097A mutation in the DH domain functionally and saw no impact on the rate of NADPH oxidation. This observation is aligned with natural amino acid variation observed at this site in metazoan FASN proteins.

L153: Fig 4: The authors indicate "entry" and "exit" with arrows for a substrate in Pig DH. How do the authors depicted such motion of the substrate that during catalysis at this active site is covalently tethered to the bulky ACP Ppant cofactor through the narrow tunnel ?

The reviewer is correct. There should not be an exit arrow as the acyl-intermediate remains covalently linked to the ACP domain during the DH-mediated dehydration reaction. We have corrected this mistake in the revised figure 4.

L163: Have the authors also considered to run 3DVA for more than 3 modes ?

Indeed, we have tried running 3DVA with more than 3 nodes, however we consistently observed three major continuous conformational changes as described in the first version of the manuscript, independent of the number of modes.

L198: Activity of KR: Compare to literature values.

The specific KR activity of purified recombinant human FASN from Sf9 cells was reported at 240 ± 40 nmol of substrate per minute per mg of protein using acetoacetyl-CoA as substrate (PMID 16215233). This is within range of the specific activity that we observed (line 253 of the revised manuscript). Please see our response below for re-calculating the initial velocity using a more optimized linear range. Additionally, we have changed the definition of the activity unit (i.e., U) from 2 μ mol of NADPH consumed per minute in the initial manuscript, to U = 1 μ mol of NADPH consumed per minute in the revised manuscript to report a consistent unit with previous measurement on human FASN KR activity (PMID 16215233). In some studies U was defined as 1 μ mol of malonyl-CoA (= 2 μ mol of NADPH) consumed per minute.

L206: "was generate at" : language

This is now changed to "was reconstructed to" in the revised manuscript.

L214: Fig 4A: The assay shows plots with unusual and difficult to reconcile jumps in the curve. The

selection of the linear range for determining the initial velocity may be affected by this. Overall the Absorption is very high, (2.2-2.6) indicating suboptimal assay conditions.

We agree with the reviewer that the velocity was not determined accurately using the first 200 seconds of measurement. We have now re-calculated specific activity using the initial slope calculated from the first 600 seconds of data that covers the observed variation in this assay and has a $R^2 > 0.98$ ($R^2 = 0.95$ for the first 200 seconds). Additionally, we have repeated the KR specific assay using FASN modifying region purified via method 2 (i.e., expressed and purified in isolation) and calculated its KR specific activity similarly by using the first 600 seconds of data ($R^2 = 0.99$). We now show KR specific assay for both method 1 and method 2 protein preparations in the revised figure 5. The variation in the error bars and the jumps in the curves, are because these data are average of two biological replicates (two independent protein purification for each curve) each with two technical replicates (i.e., four measurements for each time point for each curve). Please see our reply to reviewer 1, point 4 regarding the absorption values. In the revised manuscript we have normalized all datapoints for each assay to the value at the start of the measurement for that assay.

L214, Fig. 4C: Carved map: Appearances is of around 3.2-3.5Å local resolution, certainly not atomic resolution.

Yes, we agree with the reviewer that the density around the Denifanstat binding site appears to have worse than 3.0 Å local resolution. Please see our response above regarding map sharpening.

L247: provide yields, see above.

Yes, we agree. We have now provided yields as per reviewer's request (line 320 to 323).

L262: 8-15% agarose gel : SDS polyacrylamide gel ?

Yes, it should be SDS polyacrylamide gel. It has been corrected in the revised manuscript. We thank the reviewer for catching this error.

L270: replicate: text is not fully clear to me, is this what is meant: Purified and processed samples from two different transfections were used, and for each of those to independent assays were pipetted on different plates ?

We have modified to the statement for clearance: "More specifically, two protein preparations from two separate transfections were used (i.e., biological duplicate). For each biological replicate, two independent assays were run on the same 96-well plate (i.e., technical duplicate)." (Line 351-353). Therefore, four assays were on the same plate and the four measurements are averaged and plotted with error bars representing standard deviation between the four measurements. To run biological duplicate on the same plate at the same time, all freshly purified proteins were flash

frozen in 20 μ l aliquots in a thin-walled PCR tube using liquid nitrogen. They were thawed before the start of the reaction to synchronize the conditions between the two biological replicates.

We sincerely thank both reviewers for their valued feedbacks to improve this manuscript.

REVIEWERS' COMMENTS

Reviewer #1 (Remarks to the Author):

Thanks to the authors who have worked on most of my points in sufficient detail. The manuscript improved, but there are several comments I would like to make addressing previously raised points as well as new data and new information included during revisions. Overall, I consider this work as very relevant in providing a platform for structural studies on FAS inhibition and and modification. I

Regarding comment 1 of Reviewer 1:

It is recommended to study and cite the work of Witkowski et al. (doi:10.1021/bi048988n), who analyzed the modifying enzymatic domains of FAS in detail. This work is particularly important regarding the newly included enzymatic data to the DH domain.

Regarding comment 3 of Reviewer 1:

Can the authors explain the properties of the pcDNA3.1 vector, specifically whether the gene is over-expressed in HEK cells. The modification with PTMs, here phosphopantetheinylation is most relevant, could depend on the expression level simply because the PTM machinery (here again specifically in regard of the PPTase) may lack the capacity for large amounts of protein. The expression level of proteins is also relevant in light of the discussion about the native-like TEV cleavage approach.

line 25:

(1) „found in majority of prokaryotic and eukaryotic species“ is not an accurate statement, unless the authors cite work that has shown that archaea are underrepresented on this planet, as compared to bacteria, eukaryotes... Better state „found in bacteria and eukaryotes“

line 27:

„functional protein“ to account for ACP being non-catalytic.

line 217:

What is meant with „chemical homogeneity“?

line 193:

The authors include new data to the purification of truncated FAS, i.e. the core modifying part. They then argue that the use of the TEV cleavage approach is favorable due to the native-like situation of just having a TEV-sequence inserted that does not change the functionality of the system. I agree with most of the arguments, especially with the points that full length protein rather enables correct PTM patterns and offers the chance to trap the complex with acylated ACP. The authors further suggest that the TEV cleavage approach would be better suited to study the interaction of ACP with catalytic domains (line 224 and following). However, since domains would need to be functionally knocked-out, the protein would be non-native (similar as the truncated modifying core constructs), such that the TEV cleavage approach would not necessarily deliver the better suited protein.

line 266:

The authors now present kinetic data to the DH domains. The assay they performed is in principle ok, but its description and data analysis are sparse. There are several unclear points that should be explained: (1) The assay is performed with two types of proteins, the TEV modified full length protein and the modifying part only. While none of the modifying part variants are active, the FAS WT and the L1097A show turnover of NADPH. (2) The assay uses hydroxybutyryl-CoA as a substrate, and monitors the consumption of NADPH. The concept of this assay is not described, but it seems that the authors inspect in fact a coupled reaction of first the DH-mediated dehydration of hydroxybutyryl to crotonyl with subsequent ER-mediated reduction of the crotonyl to butyryl. The coupling of the dehydration and reduction makes sense, since the dehydration reaction is an equilibrium with an higher activity in the backward direction (see also Witkowski et al. 10.1021/bi048988n). (3) The absence of activity for the modifying region could be explained by the absence of the MAT domain, that in full length FAS can acylate the ACP with hydroxybutyryl (see Witkowski et al. 10.1021/bi048988n and Rittner et al. 2018). Thus, when using the modifying region only, just hydroxybutyryl-CoA is available and a bad substrate for the coupled reaction. However, with full length FAS, the MAT forms hydroxybutyryl-ACP which is then turned over by DH and ER. (4)

Hydroxybutyryl-CoA is the standard substrate for this purpose, and is commercially available at reasonable costs. However, given the positioning of the residue at the bottom of the binding pocket, it may just impact the processing of longer-chain substrates. In this light, the outcome of the assay is not surprising. Thus, the new data just demonstrate that both variants of DH run at similar turnover for this type of substrate, but they can not resolve a more fine tuned impact of the open vs. closed channel. I consider the DH variation as less relevant for a broad readership, and sufficiently presented in the current version of the ms (except that data collection and analysis should be explained in more detail). Additional points to the assay: (5) The x-axis should be in suitable scale. (6) Individual data points should be presented of the two biological replicates and the two technical repeats per „biological sample“ (as the SI states), giving four data points by time unit (e.g. in code x,x,o,o). (7) The authors should please translate absorbance in concentrations, such that turnover rates are received that could then be compared to overall rates of the enzyme or earlier literature. (8) The authors may also show a SDS-PAGE gel comparing the modifying parts received from the two different preparations.

line 307:

it should be clearly stated that the activity was taken from the KR domain, as, in the current version, the statement could be mixed up with the overall activity. Further, individual numbers should be given for the activity of the constructs as well as the biological samples. The authors argue that activity differences result from the purity of the different proteins; here, a SDS-PAGE gel could support the argument (see also above, points 8).

Reviewer #2 (Remarks to the Author):

The authors have addressed all issues raised by this reviewer. In particular, it is appreciated that they conducted experimental testing of the suggested method for expression of just the modifying domain and report the results in the manuscript; this will be relevant and helpful information for the interested reader.

We thank both reviewers for their positive feedbacks and effort to review and improve this manuscript. Please see our point-by-point response below. Reviewers' comments are italicized and underlined. All changes in the revised manuscript word file are tracked.

Reviewer #1 (Remarks to the Author):

Thanks to the authors who have worked on most of my points in sufficient detail. The manuscript improved, but there are several comments I would like to make addressing previously raised points as well as new data and new information included during revisions. Overall, I consider this work as very relevant in providing a platform for structural studies on FAS inhibition and and modification. I

We thank the reviewer for the favourable consideration of our manuscript.

Regarding comment 1 of Reviewer 1:

It is recommended to study and cite the work of Witkowski et al. (doi:10.1021/bi048988n), who analyzed the modifying enzymatic domains of FAS in detail. This work is particularly important regarding the newly included enzymatic data to the DH domain.

Indeed, this paper is highly relevant to the DH assay, and we have now cited this study in our revised manuscript (lines 190-192). In particular, Witkowski *et al* describe in detail the equilibrium nature of DH catalyzed reaction in mammalian FASN using hydroxybutyryl and crotonyl substrates that were either conjugated to N-acetylcysteamine or coenzyme A. We thank the reviewer for suggesting to add this reference to the revised manuscript.

Regarding comment 3 of Reviewer 1:

Can the authors explain the properties of the pcDNA3.1 vector, specifically whether the gene is over-expressed in HEK cells. The modification with PTMs, here phosphopantetheinylation is most relevant, could depend on the expression level simply because the PTM machinery (here again specifically in regard of the PPTase) may lack the capacity for large amounts of protein. The expression level of proteins is also relevant in light of the discussion about the native-like TEV cleavage approach.

This is a valid point raised by the reviewer. pcDNA3.1 harbours the human cytomegalovirus (CMV) promoter. It drives constitutive expression of genes under its control. CMV is therefore a commonly used promoter to drive over-expression of target genes in human cell lines. Therefore, overexpressed FASN may not be fully phosphopantetheinated as the endogenous PPTase activity of HEK293F cells may not have the capacity for large amount of over-expressed protein. We have mentioned this point in the revised paper (lines 195-196). To alleviate limiting PPTase activity on

over-expressed FASN, future experiments should either co-express PPTases such as Sfp or AcpS enzymes with FASN in HEK293F cells or perform the phosphopantetheinylation in vitro on using purified FASN and PPTase enzymes. To capture native PTM using the TEV cleavage method, FASN protein should be expressed at endogenous level ideally using its own native promoters and terminators as we discussed in the previous revised version of the manuscript (lines 153-155).

line 25:

(1) „found in majority of prokaryotic and eukaryotic species“ is not an accurate statement, unless the authors cite work that has shown that archaea are underrepresented on this planet, as compared to bacteria, eukaryotes... Better state „found in bacteria and eukaryotes“

The reviewer is correct, and we have modified the statement in the revised manuscript accordingly (lines 24-25).

line 27:

„functional protein“ to account for ACP being non-catalytic.

We have modified to functional protein as per reviewer's request (line 27).

line 217:

What is meant with „chemical homogeneity“?

In this statement, "chemical homogeneity" refers to protein purity. However, we realize that this description is not accurate since FASN can have heterogenous post-translational modifications (i.e., phosphopantetheinylation). In the revised statement we have removed the phrase "chemical homogeneity" (line 155). We thank the reviewer for the detailed review.

line 193:

The authors include new data to the purification of truncated FAS, i.e. the core modifying part. They then argue that the use of the TEV cleavage approach is favorable due to the native-like situation of just having a TEV-sequence inserted that does not change the functionality of the system. I agree with most of the arguments, especially with the points that full length protein rather enables correct PTM patterns and offers the chance to trap the complex with acylated ACP. The authors further suggest that the TEV cleavage approach would be better suited to study the interaction of ACP with catalytic domains (line 224 and following). However, since domains would need to be functionally knocked-out, the protein would be non-native (similar as the truncated modifying core constructs), such that the TEV cleavage approach would not necessarily deliver the better suited protein.

We thank the reviewer for agreeing with our arguments regarding the utility of TEV cleavage approach. Regarding our argument for utility of TEV cleavage approach to study ACP domain interaction with inactivated catalytic domains within the FASN core modifying region (i.e., KR, DH, and ER), we agree with the reviewer that the inactivated protein will be non-native. However, we argued that TEV cleavage method may enable acylation of holo ACP domain in cis (i.e., within one complex) using the active MAT domain and a desired CoA-ester (lines 157-169). This can then be followed by separation of the modifying region of FASN containing acylated ACP via TEV cleavage and study of ACP interactions with KR, DH, and ER using cryoEM. On the other hand working with isolated modifying region, one will need to use PPTase such as Sfp or AcpS to acylated ACP domain in trans, which will require additional protein preparations.

line 266:

The authors now present kinetic data to the DH domains. The assay they performed is in principle ok, but its description and data analysis are sparse. There are several unclear points that should be explained: (1) The assay is performed with two types of proteins, the TEV modified full length protein and the modifying part only. While none of the modifying part variants are active, the FAS WT and the L1097A show turnover of NADPH. (2) The assay uses hydroxybutyryl-CoA as a substrate, and monitors the consumption of NADPH. The concept of this assay is not described, but it seems that the authors inspect in fact a coupled reaction of first the DH-mediated dehydration of hydroxybutyryl to crotonyl with subsequent ER-mediated reduction of the crotonyl to butyryl. The coupling of the dehydration and reduction makes sense, since the dehydration reaction is an equilibrium with an higher activity in the backward direction (see also Witkowski et al. 10.1021/bi048988n). (3) The absence of activity for the modifying region could be explained by the absence of the MAT domain, that in full length FAS can acylate the ACP with hydroxybutyryl (see Witkowski et al. 10.1021/bi048988n and Rittner et al. 2018). Thus, when using the modifying region only, just hydroxybutyryl-CoA is available and a bad substrate for the coupled reaction. However, with full length FAS, the MAT forms hydroxybutyryl-ACP which is then turned over by DH and ER. (4) Hydroxybutyryl-CoA is the standard substrate for this purpose, and is commercially available at reasonable costs. However, given the positioning of the residue at the bottom of the binding pocket, it may just impact the processing of longer-chain substrates. In this light, the outcome of the assay is not surprising. Thus, the new data just demonstrate that both variants of DH run at similar turnover for this type of substrate, but they can not resolve a more fine tuned impact of the open vs. closed channel. I consider the DH variation as less relevant for a broad readership, and sufficiently presented in the current version of the ms (except that data collection and analysis should be explained in more detail). Additional points to the assay: (5) The x-axis should be in suitable scale. (6) Individual data points should be presented of the two biological replicates and the two technical repeats per „biological sample“ (as the SI states), giving four data points by time unit (e.g. in code x,x,o,o). (7) The authors should please translate absorbance in concentrations, such that turnover

rates are received that could then be compared to overall rates of the enzyme or earlier literature. (8) The authors may also show a SDS-PAGE gel comparing the modifying parts received from the two different preparations.

We thank the reviewer for in-depth analysis of the DH assay and agreeing that the assay is in principle correct. We also agree that more description of the assay, data collection and analysis should be given in the manuscript and have revised manuscript text accordingly. Specifically, we have discussed points (1) to (4) above in the revised manuscript (lines 190-200). For (5), we have chosen a scale for x-axis to demonstrate the initial rates of the reactions. For (6), we have submitted raw and processed data for the activity assays as supplementary data files. We have also shown the individual data points plotted as a new supplementary figure 10 as the reviewer requested. Please note that the DH activity assays were done as technical duplicates as described in the previous revision (Supplementary figure 4). Regarding point (7), we have translated absorbance to NADPH concentration in the new supplementary data files for each activity assay. For point (8), We have added the SDS-PAGE gels of purified FASN constructs used in this study in a new supplementary figure 9.

line 307:

it should be clearly stated that the activity was taken from the KR domain, as, in the current version, the statement could be mixed up with the overall activity. Further, individual numbers should be given for the activity of the constructs as well as the biological samples. The authors argue that activity differences result from the purity of the different proteins; here, a SDS-PAGE gel could support the argument (see also above, points 8).

We agree with the reviewer and have modified the manuscript to emphasize that the activity was taken from the KR domain only (line 227). Individual data points for the activity of each construct as well as biological samples and conversions of absorbance to NADPH concentrations are now provided in a new supplementary data file and referred to in the revised manuscript (lines 324-326). As the reviewer requested, we show the SDS-PAGE gel for two independent protein purifications for each sample used in the KR-specific activity assay in a new supplementary figure 9.

We thank the reviewer for the detailed and in-depth review that has improved our manuscript considerably.

Reviewer #2 (Remarks to the Author):

The authors have addressed all issues raised by this reviewer. In particular, it is appreciated that they conducted experimental testing of the suggested method for expression of just the modifying domain and report the results in the manuscript; this will be relevant and helpful information for the interested reader.

We thank the reviewer for the positive consideration of the revised manuscript and their effort to improve this study.